# Emotional Problems, Quality of Life and Symptom Burden in Patients with Chordoma

**DOI:** 10.3390/healthcare11081135

**Published:** 2023-04-14

**Authors:** Youtu Wu, Chuzhong Li, Huifang Zhang, Guoqin Wang, Linkai Jing, Guo Yi, Xuejun Yang, Songbai Gui, Hua Gao, Yazhuo Zhang, Guihuai Wang, Jiwei Bai

**Affiliations:** 1Neurosurgical Department, Beijing Tsinghua Changgung Hospital, Beijing 102218, China; 2Beijing Neurosurgery Institute, Capital Medical University, Beijing 100070, China; 3Department of Neurosurgery, Beijing Tiantan Hospital, Capital Medical University, Beijing 100070, China; 4China National Clinical Research Center for Neurological Diseases, Beijing Tiantan Hospital, Capital Medical University, Beijing 100070, China; 5Brain Tumor Center, Beijing Institute for Brain Disorders, Beijing 100070, China

**Keywords:** chordoma, depression, quality of life, symptom, logistic analysis

## Abstract

Chordomas are very rare malignant bone tumors. Following surgery, their effects on neurological, physical, psychological, social, and emotional functioning are substantial and can have a major impact on a patients’ quality of life (QOL). In this survey, we aimed to characterize the postoperation health-related QOL and emotional problem in patients with chordoma using the European Organization for Research and Treatment of Cancer Quality of Life Questionnaire 30 (EORTC QLQ-C30) and Hamilton Depression Rating Scale (HAMD). The cohort included 100 patients who underwent resection surgery between 2014 and 2020. Being single or divorced, living in a rural area, receiving a diagnosis of sacrococcygeal chordoma, Karnofsky performance status (KPS) ≤ 70, and weight loss were associated with increased likelihood of depression (*p* < 0.05). Patients who were single or divorced, with KPS ≤ 70, and experiencing weight loss had a higher likelihood of a worse QOL (*p* < 0.05). The uni- and multivariate logistic regression analyses indicated that the KPS level (*p* = 0.000) and postoperative radiation therapy (*p* = 0.009) were related to depression; marital status (*p* = 0.029), KPS level (*p* = 0.006), and tumor location (*p* = 0.033) were related to worse QOL. Certain characteristics placed patients with chordoma at increased risk of emotional problems, which are associated with a lowered QOL and a higher symptom burden. Further knowledge regarding emotional problems is key to improving the QOL for patients with chordoma.

## 1. Introduction

Chordoma is a rare bone cancer (0.8 cases per 1 million population/year) that is aggressive and locally invasive and has a poor prognosis [1]. Most chordomas (~95%) involve the axial skeleton, with the skull base, vertebral bodies, and sacrococcygeal bones affected in roughly equal proportions [2]. The incidence of chordoma varies by sex and race according to the United States Surveillance Epidemiology and End Results data, including more women and more people who are White, occurring earlier in life in skull base tumors [3]. Salvage therapy is challenging because of its relentless nature and refractoriness to adjuvant therapies, related to the deep-seated location of the tumor and its close proximity to critical neurovascular structures. Age at diagnosis, primary site, disease stage, surgical treatment, and tumor size are significantly associated with the prognosis of skull base chordoma [4]. Currently, chordomas are optimally managed with aggressive surgery that preserves key structures and postoperative radiation by a multidisciplinary team [5]. High-dose radiation therapy is recommended to lower the risk of recurrence [6]. To date, neurosurgeons and scientists have mainly focused on the molecular signaling pathway of and the targeted therapy for chordoma, the surgical treatment of chordoma, and the optimization of radiotherapy and immunotherapy of chordoma [7].

Cancer is often life-threatening, disfiguring, and unpredictable; hence, cancer can undermine a patient’s basic and often positive beliefs and expectations about themselves, their future, and their social relationships [8]. Patients with chordoma commonly experience multiple simultaneous symptoms of varying severity that impact their quality of life (QOL). The poor prognosis and the symptom burden resulting from chordoma therapy often bring the burden on the patients to the forefront. QOL is a multidimensional perspective that includes physical, psychological, social, and spiritual dimensions. Changes in one QOL dimension can influence perceptions in other dimensions [9]. QOL has been established as an important concept and target for research and practice in the fields of health and medicine [10]. QOL plays an important role in improving overall survival in patients with malignant tumors [11,12]. The problems revealed by patients lead to modifications and improvement in treatment and care or may show that some therapies offer little benefit. At present, there are few papers focusing on the QOL of patients with spinal chordoma [13,14]. In one study conducted in the United States, researchers found that patients with chordoma had worse anxiety and pain scores compared with the national average. Additionally, patients with recurrent or residual chordoma reported worse QOL, anxiety, and depression compared with patients with primary chordoma. However, the knowledge of postoperative emotional distress and QOL in patients with skull base chordoma is still insufficient.

In this study, we evaluated the emotional problems, QOL, and symptom burden of 100 patients with chordoma, as well as the predictive factors and correlation between emotional problems, QOL, and clinical features in these patients. The specific aim was to identify predictors of QOL, which may serve as future targets for interventions to improve QOL in this chordoma population.

## 2. Methods

### 2.1. Participants

We consecutively enrolled 100 patients with chordoma from June 2015 to June 2021. The inclusion criteria were: (1) age 18–65 years; (2) pathological results showing chordoma and brachyury (+); (3) life expectancy >12 months; (4) able to independently complete the QOL questionnaire. The exclusion criteria were: (1) severe dysphagia; (2) breast-feeding or pregnant women; (3) unstable heart disease; (4) renal failure or history of eating disorder; (5) dementia, psychosis, or impaired physical mobility; (6) history of depression. The mean follow-up period of this study was 65 (range, 15–87) months. This study was approved by the Medical Ethics Committee of Beijing Tiantan Hospital and the Medical Ethics Committee of Beijing Tsinghua Changgung Hospital. Written informed consent was obtained from the legal representatives of all the patients in accordance with the Declaration of Helsinki. All the doctor-administered clinical assessments were completed by two trained psychiatrists. Eligible candidates self-reported no current pharmacotherapy for depression at the trial screening. Additional eligibility requirements included being medically stable with no uncontrolled cardiovascular conditions; having no personal or family history (first or second degree) of psychotic or bipolar disorders; and, for women, being nonpregnant, being non-nursing, and agreeing to use contraception.

In this study, 124 patients were enrolled, including 122 baseline assessments, and 2 were found to be ineligible due to prescreen HAM-D. In total, 100 completed 12-month questionnaires were received: 8 dropped out, 13 were lost to follow-up, and 1 declined clinician assessment. The participants were not paid for their involvement in the study. The study period was from June 2015 to June 2021, which included the COVID-19 pandemic, which created considerable barriers to follow-up.

### 2.2. Emotional Assessments

The Hamilton depression rating scale (HAMD), a version of the 24-item questionnaire, was used by blinded researchers at the 12-month follow-up visit after surgery. The instrument measures somatic and affective symptoms of depression. Each item is scored for severity on a scale of 0–2 or 0–5 points, with a higher score reflecting higher symptom severity. The staff conducting the consensus diagnoses were blinded to the HAMD scores of the participants. To ensure the consistency and reliability of the scores throughout the study, two psychiatrists received training on the use of HAMD before the study. The criterion for major depressive disorder was classified as follows: no depression, score 0–7; mild, score 8–20; moderate, score 21–35; severe, score ≥36.

### 2.3. QOL Assessments

The Chinese version of the quality-of-life score 30, version 3.0, of the European Organization for Research and Treatment of Cancer was used in this study. The questionnaire was composed of 30 items related to functional status (15 questions, including physical, role, emotional, cognitive, and social functioning), global health status (2 questions), and symptoms (13 questions) regarding the respondent’s last week. The QOL was self-assessed by the patients, and the standard scores of 0 to 100 were converted using the original score for each part of the instrument. In this study, a higher score generally indicated a better status or functioning; however, a higher score on the symptom domain indicated a worse condition. Each questionnaire package began with a page of questions about demographic and health characteristics and whether the patients were receiving chemotherapy and/or radiotherapy.

### 2.4. Statistical Analyses

The measurement data are shown as means ± standard deviation (SD) or as numbers (percentage). Chi-square or Fisher’s exact tests were used for categorical data. Continuous data were evaluated as to whether they met assumptions of the parametric statistical tests. Independent sample *t*-tests were used to compare the participant characteristics and different index scores. Univariate and multivariate logistic regression analyses were performed to identify factors associated with depression or QOL. Differences correlating to an error of probability *p* < 0.05 were considered statistically significant.

## 3. Results

### 3.1. Demographic Characteristics

Of the respondents, 64 were men and 36 were women; they were aged 42.7 ± 3.7 years (range 18–64); 87 were married, and the remainder included single (6 patients) and divorced (7 patients) individuals. The educational level was: 16 patients, <6 years; 25 patients, 6–9 years; 21 patients, 10–12 years; 38 patients, >12 years. A total of 44 patients lived in urban areas, and 56 lived in rural areas. In total, we noted 64 primary and 36 recurrent tumors. The score on the Karnofsky performance status (KPS) was 84.59 ± 1.98, ranging from 30 to 100, with 33 patients scoring ≤70 and 67 patients scoring >70. Of the tumors, 21 were located in the sacrococcygeal region and 79 in the skull base, as shown in Table 1. 

### 3.2. Disease Characteristics Related to Emotional Problems

The HAMD score of the 100 patients with chordoma was 14.2 ± 2.3; 33 patients were categorized as having depression (Table 2). We observed a tendency for a gradual increase in the HAMD score with education level (F = 5.423, *p* = 0.004). The likelihood of depression was higher in patients with sacrococcygeal than in those with skull base tumors (13/21 vs. 20/79, *p* = 0.002). The likelihood of depression was higher in single or divorced patients than in married patients (8/12 vs. 25/88, *p* = 0.021), although we found no significant difference in HAMD score (*p* > 0.05). The patients with depression were more likely to have a KPS ≤ 70 (25/33 vs. 8/67, *p* = 0.000), weight loss (15/33 vs. 13/67, *p* = 0.002), or to be living in rural areas (22/33 vs. 34/67, *p* = 0.030) than those without depression. 

As shown in Table 3, the results of uni- and multivariate logistic regression analysis suggested that the KPS (B = −0.124, SE = 0.030, and *p* = 0.000) and postoperation radiation therapy (B = 2.55, SE = 0.982, *p* = 0.009) were related to depression. In addition, being single or divorced (r = 0.235, *p* = 0.019), having a low education level (r = −0.201, *p* = 0.045), being diagnosed with a sacrococcygeal tumor (r = 0.317, *p* = 0.001), and experiencing recurrence (r = 0.361, *p* = 0.000) were associated with increased likelihood of having depression and worse QOL, whereas KPS correlated with a lower likelihood of depression and a better QOL (r = 0.627, *p* = 0.000). 

### 3.3. Disease Characteristics Related to QOL and Symptoms

The self-evaluation score of chordoma patients was 51.38 ± 4.61. The highest score on the functional scales occurred for social functioning, following by role functioning and physical, and the lowest score was for emotional functioning. The patients reported that physical fatigue was the symptom of most concern, followed by appetite loss and chronic pain. 

Based on the score for QOL, the patients were divided into three groups: QOL < 40; 40 ≤ QOL ≤ 60 and QOL > 60 (Table 4). We found that there was more likelihood of QOL < 40 in single or divorced patients (*p* = 0.015), and in those with KPS < 70 (*p* = 0.016), weight loss (*p* = 0.015) or sacrococcygeal chordoma (*p* = 0.000). 

Uni- and multi-variable logistic regression analysis suggested that marital status (B = 1.584, SE = 0.723, *p* = 0.029), KPS level (B = −0.038, SE = 0.014, *p* = 0.006) and tumor location (B = 1.298, SE = 0.608, *p* = 0.033) were related to worse QOL (Table 5). In addition, there was positive correlation of QOL with KPS (r = 0.394, *p* = 0.000), and negative correlation of QOL with marital status (r = −0.218, *p* = 0.029) and tumor location (r = −0.208, *p* = 0.038). 

## 4. Discussion

Chordomas usually have the characteristics of being locally aggressive and causing chronic pain, although they grow relatively slowly and are localized [15]. In addition, clinical symptoms of chordoma depend on the lesion location [16]. Skull base chordomas often present with headaches, dysphagia or bucking caused by cranial nerve palsies, and even hemiplegia [17]. Mobile spine or sacral chordomas often result in chronic back pain or urinary/bowel dysfunction due to nerve root compression [7,18]. There is little research on postoperative management of chordomas, although they show relatively poor prognosis [19]. In this study, we focused on the emotional problems, QOL, and symptom burden in patients with chordomas. Uni- and multi-variable logistic regression analysis showed that marital status, tumor location, postoperative radiation therapy and KPS were the independent risk factors for depression or poor QOL. We hope our findings provided the opportunity to offer a tailored strategy for the negative mood of patients with chordomas.

Depression is indicated by a depressive mood, slow thinking, and loss of interest [20]. The poor recognition of depression is associated with reduced quality of life and survival [21]. Depression affects up to 20% and anxiety up to 10% of patients with cancer, compared with 5% and 7% for past-year prevalence in the general population, respectively [22]. Chordoma is a clinically and histologically unique malignant neoplasm, and treatment options have largely been centered on surgical excision, with marginal results [23]. The strategies used for diagnosis and treatment of chordomas are frequently accompanied by changes in physical status and function, unpleasant side effects, and impaired social relationships. In this study, we found that more than 30% of patients with chordoma experienced emotional problems, and depression. The likelihood of depression occurring was higher in single or divorced patients than in married patients, in those living in rural rather than urban areas, in those with KPS ≤ 70, weight loss, and sacrococcygeal rather than skull base chordomas. Patients with chordomas suffer from negative emotions under chronic psychological stress. The emotional elements of postoperation survival are commonly mentioned by patients with chordoma and their family members. This population-based, cross-sectional study assessing emotional questions among patients with chordoma showed that elderly, single or divorced patients, and patients with recurrence, KPS ≤ 70, or education <12 years reported an increased likelihood of depression. Patients with a higher level of education were less fatalistic and showed less avoidance. We suspect that somatosensory abnormalities after the completion of treatment also caused emotional problems in patients with chordoma. Unexpectedly, we found no difference in the prevalence of depression among the different age groups, and the risk of depression in patients with recurrence was not higher than that in patients with the primary disease. The results of uni- and multivariate logistic regression analyses indicated that KPS and postoperative radiation therapy were independent risk factors of depression in patients with chordoma.

A higher score for the EORTC QLQ-C30 correlated with better physical, role, emotional, and social functioning in patients with chordoma. Our findings indicated that QOL is determined by marital status, KPS, postoperative weight loss, and tumor location. Additionally, we found no significant difference in patient QOL for the different age groups and education levels. The highest score on the functional scales in patients with chordoma occurred for social functioning, following by role and physical functioning, and the lowest score was obtained for emotional functioning. Role functioning considers an individual as physical within their age and social setting [24]. Social and physical activity levels reduce after cancer diagnosis and rarely return to baseline levels following treatment completion [25]. The results of uni- and multivariate logistic regression analyses indicated that marital status, KPS, and tumor location were independent factors related to postoperative QOL in patients with chordoma.

Clinically relevant emotional problems were present in more than half of all patients with head and neck cancer and dysphagia; furthermore, there was also a significant negative correlation between the presence of aspiration and emotional problems [26]. In this study, cerebrospinal fluid leak was the most common complication after surgery, followed by neurologic disorders including dysphagia or bucking, and chronic pain. No notable improvement in preoperative symptoms occurred in patients with chordoma except for reducing the tumor compression or bulk. Furthermore, no remarkable improvement in depression or QOL was noted after postoperative radiation therapy in our study. At present, no medical therapy options are currently approved by the Food and Drug Administration for advanced chordoma, and surgical resection of chordoma has high morbidity with prolonged recovery and complications. Although this radiotherapy can reduce the tumor recurrence rate, it does not improve postoperative symptoms; therefore, the patient’s postoperative depression is not substantially improved despite the radiotherapy received. Nevertheless, we agree that systemic therapy and intensive postsurgical treatment are needed in patients with chordoma. Simultaneously, interest remains high in exploring future treatments to improve quality of life while maintaining the disease control rates of radiation and surgery combined. 

Over the most recent decade, many efforts have been devoted to prioritizing the generation of useful chordoma cell lines and tumor models, which have provided information on this malignancy and facilitated efficacious drug discovery [27,28]. At present, survivors of chordoma struggle with the after-effects, post-treatment complications, and post-traumatic stress symptoms, which can considerably diminish their quality of life. Among the 16 randomized trials of interventions targeting the caregiver, the caregiver–patient dyad, or the patient and their family for patients with advanced cancer, the most promising results showed improvements in depression resulting from early palliative care interventions [29]. Psychotherapy, medication, or combination therapy are all treatments that control and improve depressive symptoms. Other management methods, such as sleep management, exercise, and a healthy diet, are also important. Pharmacotherapy combined with psychotherapy is more effective for people with depression than pharmacotherapy alone, and psychotherapy increases adherence to pharmacotherapy. For mild depression, evidence-based studies have shown that psychotherapy should be adopted first, particularly cognitive behavioral therapy [30,31]. All these interventions have the potential to play a role in the treatment of postoperative depressive states in patients with chordoma. Patients and multidisciplinary teams should demand information related to all the treatment modalities and their side effects and should use self-care strategies to reduce them.

The peak of chordoma incidence occurs at 40–60 years, and the age of the participants in this study was relatively younger. In addition, the group included few sacrococcygeal cases and no pyramidal cases. In China, these two tumors are essentially treated by orthopedics, so multidepartment cooperation in chordoma research can provide more confirmation evidence. In addition, though binary logistic regressions may help with data presentation, notably in tables, categorization is unnecessary for statistical analysis and has some serious drawbacks compared with multivariate regression analysis. The absence of older patients (age 65–80 years) was also a problem in this study.

In this study, we aimed to investigate the postoperative emotional problems and QOL among patients with chordoma and to determine the risk factors that affect the QOL score. Certain characteristics, including being single or divorced, being diagnosed with sacrococcygeal chordoma, and having a KPS ≤ 70, place patients with chordoma at increased risk of experiencing emotional problems, which are associated with a reduced QOL and larger symptom burden. Our results may help researchers, providers, and care team members to better understand and treat the complex QOL needs in this population. Further knowledge regarding emotional problems will be key to improving the QOL of patients with chordoma.

## Figures and Tables

**Table 1 healthcare-11-01135-t001:** Characteristics of 100 patients with chordoma.

Features	Tumor Location		*p* Values
	Skull Base (*n* = 79)	Sacrococcygeal (*n* = 21)	
Sex			0.190
Male	48	16	
Female	31	5	
Age category (years)			0.155
18–59	61	13	
≥60	18	8	
Marital status			0.097
Married	71	16	
Single or divorced	8	5	
Education (years)			0.080
<6	9	7	
6–9	20	5	
9–12	20	1	
≥12	30	8	
Living area			0.540
Urban	36	8	
Rural	43	13	
KPS			0.000 ***
≤70	17	16	
>70	62	5	
Primary/recurrence			0.094
Primary	52	10	
Recurrence	25	11	
Weight loss			0.024 *
Yes	18	10	
No	61	11	
Postoperation radiation			0.651
Yes	42	10	
No	37	11	

* *p* < 0.05, *** *p* < 0.001.

**Table 2 healthcare-11-01135-t002:** Characteristics of 100 patients with chordoma.

Feature	Depression		*p* Values
	Yes (*n* = 33)	No (*n* = 67)	
Sex			0.806
Male	22	43	
Female	11	24	
Age category (years)			0.097
18–59	21	53	
≥60	12	14	
Marital status			0.019 *
Married	25	62	
Single or divorced	8	5	
Education (years)			0.247
<6	9	7	
6–9	8	17	
9–12	7	14	
≥12	9	29	
Living area			0.030 *
Urban	11	43	
Rural	22	34	
KPS			0.000 ***
≤70	25	8	
>70	8	59	
Primary/recurrence			0.348
Primary	19	45	
Recurrence	14	22	
Weight loss			0.006 **
Yes	15	13	
No	18	54	
Tumor location			
Skull base	20	59	0.002 **
Sacrococcygeal	13	8	
Postoperation radiation			0.135
Yes	21	31	
No	12	34	

** p* < 0.05, ** *p* < 0.01, *** *p* < 0.001.

**Table 3 healthcare-11-01135-t003:** Uni- and multivariate logistic analysis of factors associated with depression.

Catelog	Univariate Analysis		Multivariate Analysis	
	OR (95%CI)	*p* Value	OR (95% CI)	*p* Value
Age	0.269 (0.065–1.109)	0.069	--	--
Sex	0.369 (0.072–1.875)	0.229	--	--
Education level	0.541 (0.198–1.475)	0.23	--	--
Living area	0.513 (0.064–4.095)	0.529	--	--
Marital status	5.76 (0.754–43.97)	0.091	--	--
Disease time	1.215 (0.454–3.248)	0.699	--	--
Tumor location	0.862 (0.114–6.535)	0.886	--	--
Weight loss *	0.392 (105–1.473)	0.166	--	--
Radiation therapy	9.427 (1.331–66.79)	0.025	10.861 (2.122–55.60)	0.004
KPS	0.859(0.798–0.926)	0.000	0.884 (0.841–0.930)	0
Primary/Recurrence	1.896 (0.503–7.145)	0.345	--	--

* defined as more than 5% weight loss.

**Table 4 healthcare-11-01135-t004:** Associations between patient characteristics and QOL.

Features	QOL			*p* Values
	<40 (*n* = 22)	40–60 (*n* = 49)	>60 (*n* = 29)	
Gender				0.262
Male	15	34	15	
Female	7	15	14	
Age category				0.780
18–59	15	37	22	
≥60	7	12	7	
Marital status				0.015 *
Married	15	46	26	
Single or Divorced	7	3	3	
Education (year)				0.228
<6	7	5	4	
6–9	6	9	10	
9–12	3	15	3	
≥12	6	20	12	
Living area				0.401
Urban	7	24	13	
Rural	15	25	16	
KPS				0.016 *
≤70	10	9	4	
>70	12	40	25	
Primary/Recurrence				0.788
Primary	14	30	20	
Recurrence	8	19	9	
Weight loss				0.016 *
Yes	11	13	4	
No	11	36	25	
Tumor location				
Skull base	11	4	6	0.000 ***
Sacrococcygeal	11	45	23	
Postoperation radiation				0.411
Yes	14	26	13	
No	8	23	16	

* *p* < 0.05, *** *p* < 0.001.

**Table 5 healthcare-11-01135-t005:** Uni- and multi-variable logistic analysis of factors associated with QOL.

	Univariate Analysis		Multivariate Analysis	
	OR (95%CI)	*p* Value	OR (95%CI)	*p* Value
Age	0.559 (0.204–1.534)	0.259	--	--
Gender	0.868 (0.245–3.073)	0.826	--	--
Education level	0.637 (0.284–1.428)	0.273	--	--
Living area	1.045 (0.177–6.159)	0.961	--	--
Marital status	4.904 (1.069–22.50)	0.041 *	4.875 (1.181–20.13)	0.029 *
Disease time	0.624 (0.261–1.494)	0.290	--	--
Tumor location	4.104 (1.032–16.32)	0.045 *	3.662 (1.112–12.05)	0.033 *
Weight loss	0.736 (0.285–1.899)	0.526	--	--
Radiation therapy	1.896 (0.514–6.990)	0.337	--	--
KPS	0.953(0.921–0.985)	0.005 **	0.963 (0.937–0.989)	0.006 **
Primary/Recurrence	0.854 (0.233–3.129)	0.811	--	--

* *p* < 0.05, ** *p* < 0.01.

## Data Availability

The data presented in this study are available on request from the corresponding author.

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
