# Peer review of "Emotional Problems, Quality of Life and Symptom Burden in Patients with Chordoma"

_healthcare, 2023, doi:10.3390/healthcare11081135_

Round 1
Reviewer 1 Report
Introduction
Authors should consider providing global statistics, as well as country specific statistics to bolster their arguments about the prevalence and impact of chordomas. They should also consider providing information about the areas that are commonly impacted by chordomas (e.g., skull, spine, etc).
Authors state “Patients should demand information related to all the treatment modalities and their side effects and use self-care strategies to reduce them.” While this is an important statement, it does not fit with the introduction of a disease state. This statement should be discussed as part of the clinical implications in the discussion section.
There appears to be a citation problem in page 2: “Problems revealed by patients {"citationID":"wpOFzyr lead to modifications.”
Authors state that “there are a few papers focusing on the QOL of patients with spinal chordoma(9,10), and the knowledge of postoperative emotional distress and QOL in patients with skull base chordoma is still deficient.” Authors should briefly describe the outcomes of these studies and the areas that are still unknown.
Authors should consider discussing patient quality of life and emotional adjustment through the lens of a guiding theoretical approach. For example, they could use the guidance of the “Social-Cognitive Processing Theory” by Lepore (2001) which would strengthen their introduction and discussion, and guide their interpretation of the findings.
Lepore SJ (2001) A social-cognitive processing model of emotional adjustment to cancer. In: Baum A, Andersen BL (eds) Psychosocial interventions for cancer. American Psychological Association, Washington, pp 99–118
Lepore SJ, Revenson TA (2007) Social constraints on disclosure and adjustment to cancer. Soc Personal Psychol Compass 1:313–333. https://doi.org/10.1111/j.1751-9004.2007.00013.x
Methods
Authors should describe the study design and timeline clearly.
Authors should describe how patients were identified and approached for this study. Authors should also state the number of people who declined to participate. How many people were excluded? How many participants provided incomplete data? How much missingness was considered incomplete data? Were the participants paid for study participation?
Authors should consider providing example items for QOL and HAMD measures.
Authors should provide references for QOL and HAMD measures and provide information about their reliability/validity in this population.
Authors should describe Karnofsky Performance status measure and the “disease characteristics related to QOL and symptoms” measures as part of the measures, provide information about the reliability/validity of these measures in this population, as well as any cut-off points. It is unclear how the authors were able to get information about fatigue, pain, appetite loss, anxiety or other symptoms.
If relevant data are available, authors should report if patients had comorbid chronic conditions.
Authors should clarify how they categorized weight loss – for instance how much weight loss (in kg or lbs) would be considered weight loss for this study and what is the timeline for this weight loss (how much time after surgery or other assessments).
Results
From Tables 3 and 5, it appears that authors only included variables that had significant results in univariate analyses to the multivariate analyses. This approach is misleading as variables may have hidden or masked impacts that are only seen when controlling the impact of others. Thus, authors should consider re-running the multivariate analyses and include all theoretically or empirically relevant variables in the analyses (instead of only including the ones that are significant in the univariate analyses).
Of major concern, the authors should analyze the continuous HAMD and QOL scores instead of diving these into groups. This is of major concern as some of the variables have very limited number of people in specific categories leading to unbalanced groupings (e.g., very few people in the divorced/single category). Categorization of continuous variables leads to loss of information, may significantly underestimate the variability, and ultimately may lead to problematic conclusions (Altman & Royston, 2006). Similarly, if data is available, authors should analyze age and education as continuous variables.
Altman DG, Royston P. (2006). The cost of dichotomising continuous variables. BMJ. 332(7549):1080. https://doi.org/10.1136/bmj.332.7549.1080.
Discussion
Discussion section would benefit from reframing the arguments by focusing on a theoretical approach.
Of major concern, authors need to think about their findings, and thoroughly discuss potential reasons for them. In its current state the discussion section is very poorly written and continuously makes claims outside the scope of this work. It is very difficult to understand whether authors are describing their own results or results of others as many of the references are missing. Finally, restating results is not enough for discussion. Authors are highly encouraged to consider the meaning of their findings, provide alternative explanations for their results considering the literature and provide directions for future studies.
Authors need to provide references for their statements. Please review and add references throughout the discussion. A few examples that need references:
“Chordomas usually have the characteristics of being locally aggressive and causing chronic pain, although they grow relatively slowly and are localized.”
“In addition, clinical symptoms of chordoma depend on the lesion location.”
“Skull base chordomas often present with headache, dysphagia or bucking caused by cranial nerve palsies, and even hemiplegia.”
“Mobile spine or sacral chordomas often result in chronic back pain or urinary/bowel dysfunction due to nerve root compression.”
“There is little research on postoperative management of chordomas, although they show relatively poor prognosis”
“Depression is indicated by depressive mood, slow thinking, and loss of interest.” “Poor recognition of depression is associated with reduced quality of life and survival.”
Authors should clarify and re-think about their claims about their findings- while they measure QOL and depression (depressive symptoms), they do not have a similarly validated measure of anxiety. Thus, the statements below are not supported by their results:
“In this study, we found that more than 30% of patients with chordoma had emotional problems, including anxiety and depression.”
“This population-based, cross-sectional study assessing emotional questions among patients with chordoma showed that elderly, single or divorced patients, and patients with recurrence, KPS≦70 or education for <12 years reported more likelihood of anxiety or depression”
Similarly, authors did not assess coping. Thus, the statements below are also not supported by their results:
“Patients with a higher level of education were less fatalistic and showed less avoidance. There was a good correlation between anxiety or depression and pre-operation status, age, quality of life and coping.”
Authors should consider the context of their findings. This study was conducted in China. Do the authors believe the findings would be similar in Europe, USA, or across the world?
Throughout the manuscript authors make causal arguments about correlational data, which is beyond its scope and not supported by their results. They should explain the correlational findings and perhaps make suggestions for future longitudinal research to explore these relations further. It would be more helpful if authors could suggest potential mediators and moderators of the relations between caregiving and quality of life using a theoretical framework.
The authors should elaborate on the clinical, research and healthcare policy implications of these findings.
Authors should discuss the limitations of these findings.
Author Response
Queation 1.Introduction
Response: Thanks for the suggestions.
Most chordomas (~95%) involve the axial skeleton, with the skull base, vertebral bodies, and sacrococcygeal bones affected in roughly equal proportions. The incidence of chordoma varies by gender and race by the United States Surveillance Epidemiology and End Results data.
To date, neurosurgeons and scientists mainly focused molecular signaling pathway and targeted therapy of chordoma, surgical treatment of chordoma, optimization of radiotherapy, and immunotherapy of chordoma.
Caner is often life threatening, disfiguring, and unpredictable; hence, cancer can undermine patient's basic and often positive beliefs and expectations about themselves, their future, and social relationships.
We have strengthened the introduction and discussion and guided the interpretation of the findings, and have cited references in the revised manuscript as suggested by the reviewers.
Queation 2.Methods
Response: Thanks for the suggestions.
In this study, there were 124 enrolled patients, including 122 baseline assessment and 2 found to be ineligible due to prescreen HAMD. However, there were 100 completed 12-months questionnaires. There were 8 dropped out, and 13 lost to follow-up, and 1 declined clinician assessment.
The period of study was from June 2015 to June 2021 which experienced the epidemic of Covid19. The Covid19 pandemic brought the great obstacles to follow-up.
We have added the details into the revised manuscript.
Eligible candidates self-reported no current pharmacotherapy for depression at trial screening. Additional eligibility requirements included being medically stable with no uncontrolled cardiovascular conditions; having no personal or family history (first or second degree) of psychotic or bipolar disorders; and, for women, being nonpregnant, being non-nursing, and agreeing to use contraception.
The instrument measures somatic and affective symptoms of depression. Each item is scored for severity on a scale of 0-2 or 0-5 points, with a higher score reflecting higher symptom severity. The staff conducting the consensus diagnoses were blinded to the HAM-D scores of the participants. In order to ensure the consistency and reliability of the scores throughout the study, two psychiatrists received training on the use of HAMD before the study. The criteria for a major depressive disorder.
The patients had no comorbid chronic conditions. The exclusion criteria were: 1) severe dysphagia; 2) breast feeding or pregnant women; 3) unstable heart disease; 4) renal failure, history of eating disorder; 5) dementia, psychosis, impaired physical mobility; 6) history of depression.
*Weight loss: defined as more than 5 percent weight loss post-operation
We have added the sentences into the revised manuscript.
Queation 3. Results
Response: Thanks for the suggestions.
Logistic regression analysis is used to study the influence of X on Y, and there is no requirement on the data type of X. X can be classified data or quantitative data, but Y must be classified data, and the corresponding data analysis method is used according to the number of options of Y. Binary logistic regression requires that the dependent variables can only be 2 terms, and the numbers must be 0 and 1. Hence, the continuous HAMD and QOL scores shuold be cincert to groups. In this study, age, gender, education level, living area, marital status, KPS scores, primary/recurrence, weight loss and postoperaton radiation were analyzed in univariate analyses and multivariate analyses. The cut-off P value for multivariate analyses was 0.1, and remove value was 0.2 based on univariate analyses in IBM SPSS Statistics.
Older cancer patients face a unique set of age-associated changes, comorbidities and circumstances that impact on their quality of life in ways that are different from those affecting younger patients (Ann Oncol , 2018,29(8):1718-1726). Authors think scientists should pay more attention to the disease characteristics in older age group than continuous scale of age.
Queation 4. Discussion
Response: Thanks for the suggestions.
We havd added the corresponding references into the revised manuscript according to the reviewer’s suggestion.
Over the past decade, many efforts have been made to prioritize the generation of useful chordoma cell lines, and tumor models that have shed more light on this malignancy and have made efficacious drug discovery a greater possibility. At present, even survivors of chordoma struggle with late effects, post-treatment complications, and post-traumatic stress symptoms that can significantly diminish their quality of life. Among the 16 randomized trials of interventions targeting the caregiver, the caregiver-patient dyad, or the patient and their family for patients with advanced cancer, the most promising results showed improvement of depression resulting from early palliative care interventions. It would be helpful that potential mediators and moderators of the relations between caregiving and quality of life.Patients should demand information related to all the treatment modalities and their side effects and use self-care strategies to reduce them.
The limitation of the manuscript:
The peak of chordoma incidence is 40-60, and the age of the participants in this study is relatively younger. The conclusions of this paper that are independent of age cannot fully explain the occurrence of anxiety and depression in the age group of 50-60. In addition, there were few sacrococcygeal cases and no pyramidal cases. In China, these two parts of tumors are basically completed by the department of orthopedics, so multi-department cooperation in chordoma research can provide more confirmation evidence.
We have added the sentences into the revised manuscript.
Reviewer 2 Report
This study focuses on emotional problems and quality of life in patients with chordoma. It develops also model on possible risk factors.
The topic could be interesting, even if there are several limits of the current form have to be solved.
Introduction
The introduction is really scarce, it could be added more background on this topic and also the possible risk factors evidenced in the literature, so to introduce better the study aims.
Also the research questions/hypotheses are not well developed and they should be introduced by studies on this topic. You should stress more the value added of your manuscript.
Instruments
HAMD was not shown well, it should describe more, adding some examples and reporting the reliability of the questionnaire and its scales. The same thing for the QoL instrument.
Procedure
All the patients accepted to participate to this study? Were there dropouts or not? What is the period of data collection? Is before or after Covid19 pandemic? It should be added.
Results/Discussion
It should be checked if these patients had or not a psychological assistance or if they were taken psychiatric drugs or a mood therapy. This could be an important variable to add to the models, especially for emotional problems.
If this information was not collected it is important to put this as a recommendation for future research.
Strengths and limits of this study should be evidenced. In the limits it should be discussed also the not homogeneity of the sample along age ranges or gender that could influence the models.
Author Response
Queation 1. The introduction is really scarce, it could be added more background on this topic and also the possible risk factors evidenced in the literature, so to introduce better the study aims.
Also the research questions/hypotheses are not well developed and they should be introduced by studies on this topic. You should stress more the value added of your manuscript.
Response:
Thanks for the suggestion. We have added more background in the revised manuscript according to the reviewer’s suggestion .
Queation 2. HAMD was not shown well, it should describe more, adding some examples and reporting the reliability of the questionnaire and its scales. The same thing for the QoL instrument.
Response: Thanks for the sugesstion.
We have added the details into the revised manuscript.
Queation 3. All the patients accepted to participate to this study? Were there dropouts or not? What is the period of data collection? Is before or after Covid19 pandemic? It should be added.
Response: Thanks for the sugesstion.
In this study, there were 124 enrolled patients, including 122 baseline assessment and 2 found to be ineligible due to prescreen HAMD. However, there were 100 completed 12-months questionnaires. There were 8 dropped out, and 13 lost to follow-up, and 1 declined clinician assessment.
The period of study was from June 2015 to June 2021 which experienced the epidemic of Covid19. The Covid19 pandemic brought the great obstacles to follow-up.
We have added the details into the revised manuscript.
Queation 4.It should be checked if these patients had or not a psychological assistance or if they were taken psychiatric drugs or a mood therapy. This could be an important variable to add to the models, especially for emotional problems.
If this information was not collected it is important to put this as a recommendation for future research.
Strengths and limits of this study should be evidenced. In the limits it should be discussed also the not homogeneity of the sample along age ranges or gender that could influence the models.
Response: Thanks for the sugesstion.
Eligible candidates self-reported no current pharmacotherapy for depression at trial screening. Additional eligibility requirements included being medically stable with no uncontrolled cardiovascular conditions; having no personal or family history (first or second degree) of psychotic or bipolar disorders; and, for women, being nonpregnant, being non-nursing, and agreeing to use contraception.
The peak of chordoma incidence is 40-60, and the age of the participants in this study is relatively younger. The conclusions of this paper that are independent of age cannot fully explain the occurrence of anxiety and depression in the age group of 50-60.
In this study, there were few sacrococcygeal cases and no pyramidal cases. In China, these two parts of tumors are basically completed by the department of orthopedics, so multi-department cooperation in chordoma research can provide more confirmation evidence.
We have added the details into the revised manuscript.
Reviewer 3 Report
Well written
please mention ethical clearance number and date
please check grammar throughout the manuscript
Author Response
Thanks for the generous assessment of our work!
Round 2
Reviewer 1 Report
Introduction
Authors state: “The incidence of chordoma varies by gender and race by the United
States Surveillance Epidemiology and End Results data[3].” This statement does not provide any information to the reader. Please revise and describe if there are specific groups (e.g., older adults, males, etc.) who have greater prevalence of chordomas.
Please fix the typo “Caner is often life threatening…” as “Cancer is often …”
Please fix the citation problem on page 2: “Problems revealed by patients {"citationID":"wpOFzyr lead to modifications.”
Authors should revise the statement: “The score of anxiety and pain in patient with chordoma were also worse than the national average in the United States, and there were worse QOL, more anxiety and depression in recurrent or residual chorodma than primary chordoma” as: “One study conducted in the United States found that patients with chordoma had worse anxiety and pain scores compared to the national average. Additionally, patients with recurrent or residual chordoma reported worse QOL, anxiety, and depression compared to patients with primary chordoma.”
Methods
Authors should describe how patients were identified and approached for this study. Authors should also state the number of people who declined to participate. How many people were excluded? How many participants provided incomplete data? How much missingness was considered incomplete data? Were the participants paid for study participation? All of these should be reported in the paper.
Authors should provide references for QOL measure and provide information about its reliability/validity in this population.
I have stated this before: Authors should describe Karnofsky Performance status measure and the “disease characteristics related to QOL and symptoms” measures as part of the measures, provide information about the reliability/validity of these measures in this population, as well as any cut-off points. It is unclear how the authors were able to get information about fatigue, pain, appetite loss, anxiety or other symptoms.
Results
As a statistical expert, I understand what logistic regression is. I do not understand why the authors chose to run logistic regressions. They should be running multivariate regression analysis and use the continuous HAMD and QOL scores instead of artificially diving these into groups. This is of major concern as some of the variables have very limited number of people in specific categories leading to unbalanced groupings (e.g., very few people in the divorced/single category). Categorization of continuous variables leads to loss of information, may significantly underestimate the variability, and ultimately may lead to problematic conclusions (Altman & Royston, 2006). At the very least, if authors continue to use this wrong method of analysis, they should acknowledge that this limitation in their paper.
Altman DG, Royston P. (2006). The cost of dichotomising continuous variables. BMJ. 332(7549):1080. https://doi.org/10.1136/bmj.332.7549.1080.
Authors state that “scientists should pay more attention to the disease characteristics in older age group than continuous scale of age.” While this may be true, in the current study, the older age group (60-64 years looking at the range they report) consists of a very small subgroup (26 out of 100 patients). Thus, categorization of this small group for the analyses is not appropriate.
Discussion
It appears that authors did not change much in the discussion section. It is poorly written and still makes claims outside the scope of this work.
I have stated this before: Authors should clarify and re-think about their claims about their findings- while they measure QOL and depression (depressive symptoms), they do not have a validated measure of anxiety. Thus, the statements below are not supported by their results:
“In this study, we found that more than 30% of patients with chordoma had emotional problems, including anxiety and depression.”
“This population-based, cross-sectional study assessing emotional questions among patients with chordoma showed that elderly, single or divorced patients, and patients with recurrence, KPS≦70 or education for <12 years reported more likelihood of anxiety or depression”
I have stated this before as well: Authors did not assess coping. Thus, the statements below are also not supported by their results:
“Patients with a higher level of education were less fatalistic and showed less avoidance. There was a good correlation between anxiety or depression and pre-operation status, age, quality of life and coping.”
Either authors need to delete these statements or revise them and make claims within the scope of their findings.
I have also stated this before: Throughout the manuscript authors make causal arguments about correlational data, which is beyond its scope and not supported by their results. They should explain the correlational findings and perhaps make suggestions for future longitudinal research to explore these relations further. It would be more helpful if authors could suggest potential mediators and moderators of the relations between caregiving and quality of life using a theoretical framework.
Author Response
Comments and Suggestions for Authors
- Introduction
1.1 Authors state: “The incidence of chordoma varies by gender and race by the United States Surveillance Epidemiology and End Results data[3].” This statement does not provide any information to the reader. Please revise and describe if there are specific groups (e.g., older adults, males, etc.) who have greater prevalence of chordomas.
Response: Thanks for the suggestion.
The incidence of chordoma varies by gender and race by the United States Surveillance Epidemiology and End Results data, including more females, and white, and earlier life in skull base tumors.
We have re-edited the sentence in the revised manuscript.
1.2 Please fix the typo “Caner is often life threatening…” as “Cancer is often …”
Please fix the citation problem on page 2: “Problems revealed by patients {"citationID":"wpOFzyr lead to modifications.”
Response: Thanks for the suggestion.
We have corrected the writing errors in the manuscript.
1.3 Authors should revise the statement: “The score of anxiety and pain in patient with chordoma were also worse than the national average in the United States, and there were worse QOL, more anxiety and depression in recurrent or residual chorodma than primary chordoma” as: “One study conducted in the United States found that patients with chordoma had worse anxiety and pain scores compared to the national average. Additionally, patients with recurrent or residual chordoma reported worse QOL, anxiety, and depression compared to patients with primary chordoma.”
Response: Thanks for the suggestion.
We have re-edited the sentence in the revised manuscript according to the reviewer’s suggestion.
- Methods
2.1 Authors should describe how patients were identified and approached for this study. Authors should also state the number of people who declined to participate. How many people were excluded? How many participants provided incomplete data? How much missingness was considered incomplete data? Were the participants paid for study participation? All of these should be reported in the paper.
Response: Thanks for the suggestion.
In this study, there were 124 enrolled patients, including 122 baseline assessment and 2 found to be ineligible due to prescreen HAMD. However, there were 100 completed 12-months questionnaires. There were 8 dropped out, and 13 lost to follow-up, and 1 declined clinician assessment. There was no paid for the participants. The period of study was from June 2015 to June 2021 which experienced the epidemic of Covid19. The Covid19 pandemic brought the great obstacles to follow-up.
We have added the sentences into the revised manuscript.
2.2 Authors should provide references for QOL measure and provide information about its reliability/validity in this population.
Response: Thanks for the suggestion.
We apologize for the lack of reliability/validity in this study. In fact, this manuscript is our team's first paper on quality of life and psychological status after chordoma surgery, and it has many flaws and weaknesses. Two scales will be used in future studies to provide reliability/validity in this population.
2.3 I have stated this before: Authors should describe Karnofsky Performance status measure and the “disease characteristics related to QOL and symptoms” measures as part of the measures, provide information about the reliability/validity of these measures in this population, as well as any cut-off points. It is unclear how the authors were able to get information about fatigue, pain, appetite loss, anxiety or other symptoms.
Response: Thanks for the suggestion.
The score of Karnofsky Performance status (KPA) was 84.59±1.98, range 30 to 100. There were 33 patients ≤70; 67 patients >70. We have added the sentence into the revised manuscript.
The degrees of fatigue, pain, appetite loss or other symptoms were judged by the participates themselves compared with the pre-operation. For example, fatigue: 1) barely eat; 2) no more half eat; 3) half eatï¼›4)less normal eat; 5) normal eat.
3 Results
3.1 As a statistical expert, I understand what logistic regression is. I do not understand why the authors chose to run logistic regressions. They should be running multivariate regression analysis and use the continuous HAMD and QOL scores instead of artificially diving these into groups. This is of major concern as some of the variables have very limited number of people in specific categories leading to unbalanced groupings (e.g., very few people in the divorced/single category). Categorization of continuous variables leads to loss of information, may significantly underestimate the variability, and ultimately may lead to problematic conclusions (Altman & Royston, 2006). At the very least, if authors continue to use this wrong method of analysis, they should acknowledge that this limitation in their paper.
Altman DG, Royston P. (2006). The cost of dichotomising continuous variables. BMJ. 332(7549):1080. https://doi.org/10.1136/bmj.332.7549.1080.
Response: Thanks for the suggestion.
We do apologize for the statistical inadequacy in this study. In addition, though Binary logistic regressions may help data presentation, notably in tables, categorisation is unnecessary for statistical analysis and it has some serious drawbacks compared with multivariate regression analysis.
We have added the sentence into the discussion of revised manuscript.
3.2 Authors state that “scientists should pay more attention to the disease characteristics in older age group than continuous scale of age.” While this may be true, in the current study, the older age group (60-64 years looking at the range they report) consists of a very small subgroup (26 out of 100 patients). Thus, categorization of this small group for the analyses is not appropriate.
Response: Thanks for the suggestion.
The average year in skull base chordoma was 47.4 years base 664 patients (World Neurosurg. 2018 May;113:e618-e627).
The absence of old patients (age 65-80) also was problems and the pity.. We have added it into limitation of manuscript.
4 Discussion
4.1 It appears that authors did not change much in the discussion section. It is poorly written and still makes claims outside the scope of this work.
I have stated this before: Authors should clarify and re-think about their claims about their findings- while they measure QOL and depression (depressive symptoms), they do not have a validated measure of anxiety. Thus, the statements below are not supported by their results:
“In this study, we found that more than 30% of patients with chordoma had emotional problems, including anxiety and depression.”
“This population-based, cross-sectional study assessing emotional questions among patients with chordoma showed that elderly, single or divorced patients, and patients with recurrence, KPS≦70 or education for <12 years reported more likelihood of anxiety or depression”
Response: Thanks for the suggestion
In this study, we used the Hamilton Depression Rating Scale for depression, and we have deleted the sentences related to anxiety in the revised manuscript.
4.2 I have stated this before as well: Authors did not assess coping. Thus, the statements below are also not supported by their results:
“Patients with a higher level of education were less fatalistic and showed less avoidance. There was a good correlation between anxiety or depression and pre-operation status, age, quality of life and coping.”
Either authors need to delete these statements or revise them and make claims within the scope of their findings.
Response: Thanks for the suggestion
We have deleted the sentence in the revised manuscipt according the reviewer’s suggestion.
4.3 I have also stated this before: Throughout the manuscript authors make causal arguments about correlational data, which is beyond its scope and not supported by their results. They should explain the correlational findings and perhaps make suggestions for future longitudinal research to explore these relations further. It would be more helpful if authors could suggest potential mediators and moderators of the relations between caregiving and quality of life using a theoretical framework.
Response: Thanks for the suggestion
Psychotherapy, medication, or combination therapy are all treatments that control and improve depressive symptoms. Other management methods, such as sleep management, exercise, and a healthy diet, are also important. Pharmacotherapy combined with psychotherapy has a better effect on people with depression than pharmacotherapy alone, and psychotherapy increases adherence to pharmacotherapy. For mild depression, there is also evidence-based evidence that psychotherapy should be adopted first, particularly cognitive behavioural therapy(1). All of these interventions have the potential to play a role in the treatment of postoperative depressive states in patients with chordoma.
We have added the sentences into the revised manuscript.
Reviewer 2 Report
I think that the paper is really ameliorated following the reviewers' comments and suggestions. I think that now the paper could be publishable.
Author Response
Thanks for the comments of the reviewer.